# Characterization and Functional Analysis of Toll Receptor Genes during Antibacterial Immunity in the Green Peach Aphid *Myzus persicae* (Sulzer)

**DOI:** 10.3390/insects14030275

**Published:** 2023-03-09

**Authors:** Li He, Chao Zhang, Hong Yang, Bo Ding, Han-Zhi Yang, Sen-Wen Zhang

**Affiliations:** Guizhou Provincial Key Laboratory for Agricultural Pest Management of Mountainous Region, Institute of Entomology, Guizhou University, Guiyang 550025, China

**Keywords:** *Myzus persicae*, innate immunity, Toll receptor, bacterial infection, RNA interference

## Abstract

**Simple Summary:**

The Toll receptor and Toll signaling pathway are best known for their universal function in insect development and innate immunity. In this study, the effect of targeted silencing of Toll receptor genes on the resistance of the green peach aphid, *Myzus persicae* (Sulzer), to bacterial infection was evaluated. We found that the expressions of MpToll genes were significantly increased to varying degrees after bacterial treatment. Moreover, silencing the target genes and then infecting them with bacteria significantly increased the mortality of *M. persicae*. This study contributes to the continued understanding of the antimicrobial immune mechanisms of aphids and may provide guidance for developing new pesticides that can inhibit the immune system of *M. persicae*.

**Abstract:**

The insect Toll receptor is one of the key members of the Toll signaling pathway, which plays an indispensable role in insect resistance to pathogen infection. Herein, we cloned and characterized five Toll receptor genes from *Myzus persicae* (Sulzer), which were found to be highly expressed in the first-instar nymphs and adults (both wingless and winged) at different developmental stages. Expressions of MpToll genes were highest in the head, followed by the epidermis. High transcription levels were also found in embryos. Expressions of these genes showed different degrees of positive responses to infection by *Escherichia coli* and *Staphylococcus aureus*. The expression of *MpToll6*-*1* and *MpToll7* significantly increased after infection with *E. coli*, whereas the expression of *MpToll*, *MpToll6*, *MpToll6*-*1*, and *MpTollo* continuously increased after infection with *S. aureus*. RNA interference-mediated suppressed expression of these genes resulted in a significant increase in the mortality of *M. persicae* after infection with the two bacterial species compared with that in the control group. These results suggest that MpToll genes play vital roles in the defense response of *M. persicae* against bacteria.

## 1. Introduction

Insects lack the adaptive immunity found in mammals, and, therefore, their innate immune system comprising humoral and cellular immune responses is an essential host defense against pathogens [1]. These responses mainly rely on the activation of Toll, immune deficiency (IMD), c-Jun N-terminal kinase (JNK), and Janus kinase/signal transducers and activators of transcription (JAK/STAT) signaling pathways as well as melanization reactions mediated by various protein factors and responses generated by antimicrobial peptides (AMPs), lysozyme, and lectins, which have been demonstrated in *Drosophila* [2,3]. Toll and Toll-like receptors (TLRs) are an ancient group of proteins, the latter having extensive pathogen recognition capabilities in mammals [4]. The first Toll was originally defined in *Drosophila melanogaster* and was found to be a crucial protein required for dorsal-ventral embryonic polarity [5]. Mutations in the *Toll* gene were later found to significantly reduce the survival rate after fungal infection [6]. Previous studies generally agree that the activation of Toll signaling pathways depends on pattern recognition receptors (PRRs), such as peptidoglycan recognition proteins (PGRPs) and gram-negative bacteria-binding proteins (GNBPs) for pathogen recognition [7,8]. Toll receptors can bind to the endogenous ligand Spätzle cytokines to induce nuclear translocation of the NF-kB transcription factors Dorsal/Dif which subsequently regulate the expression of key innate immune effector genes in most insects [9,10,11]. A recent study suggested that individual Toll proteins in insects can recognize bacterial lipopolysaccharide through interaction with two MD-2–like proteins to activate the pathway in a similar manner to that of mammalian TLRs, which function as PRRs that directly bind to the specific pathogen-associated molecular patterns (PAMPs) [12].

Toll in *Drosophila* was first cloned and characterized by Hashimoto et al. [13]. A typical Toll protein, categorized as a type I integral transmembrane protein, is regarded as the link between extracellular and intracellular signals [14]. Toll contains an ectodomain rich in leucine-rich repeats (LRRs) and a conserved cytoplasmic Toll-IL-1R (TIR) domain that has homology to the corresponding region of the interleukin-1 receptor (L-1R) [14,15]. Currently, a large number of Toll-related genes have been found in the genomes of a wide range of insect orders, including Diptera, Hymenoptera, Lepidoptera, and Coleoptera [16,17,18,19,20]; by contrast, 10–13 TLRs have been identified in mammals [21]. It is interesting that TLRs in the Atlantic cod, *Gadus morhua* L., have been diversified as compensation for their deficiency in adaptive immunity [22]. Only some of these known insect Toll proteins have been shown to be involved in the immune response of the organism while others are known to be involved in insect development. Among the nine Toll members of *Drosophila*, *Toll*, *18w*, *Toll-5*, and *Toll-9* are known to respond to fungi or bacterial infection [23,24,25], and *Toll-6* to *Toll-8* (*Tollo*) are required for development [11]. Moreover, *AgToll* may be crucial for antifungal innate immunity and in the embryonic development process, as *AgToll9* was involved in anti*-Plasmodium* responses in *Anopheles gambiae* [26].

The green peach aphid, *Myzus persicae* (Sulzer) (Hemiptera: Aphidiade), is a major global pest, and >400 various plant species are believed to serve as hosts [27], while >100 different plant viruses can be transmitted through its direct feeding [28]. This expansive feeding behavior of aphids makes them highly destructive pests of a large amount of vital crop species [29]. Strategies combining RNA interference (RNAi) and microbial invasion have been extensively studied to control aphid populations based on the destruction of innate immunity of insects [8,30], and enriched knowledge of insect immunity may help in the search for new control targets [31]. However, relatively few studies have been conducted on *M*. *persicae* immunity, and the number and function of *M*. *persicae* Toll receptor genes remain unclear. Therefore, in this study, five MpToll genes were identified, their spatiotemporal expression profiles were characterized, and their role in *M. persicae* responses to bacteria stress was examined using microbial infection and RNAi. Our results can provide a reference for the development of suitable targets for the sustained control of *M. persicae* based on the RNAi suppression of its immune system and on microbial infection.

## 2. Materials and Methods

### 2.1. Insect Rearing and Collection

A green peach aphid colony was originally collected from the Jinzhu town test base, in Guizhou Province, China in 2019. This was subsequently maintained on young tobacco seedlings (cultivar “Yunyan 87”) in an artificial climate chamber at 25 °C ± 1 °C, 70% ± 5% relative humidity with a light: dark photoperiod of 14: 10 h, and without exposure to insecticides. To harvest synchronously developed aphids, parthenogenetic wingless adults were placed on new seedlings for reproduction and removed after 24 h; newborn nymphs were reared sequentially to the corresponding stage for experiments.

### 2.2. Characterization and Sequencing of M. persicae Toll Receptor Genes

The MpToll genes were identified by searching the TIR domain in the genome (GenBank accession: PRJNA296778) [29] and the transcriptomic database of our laboratory (GenBank accession: PRJNA412304). The open reading frame (ORF) sequences of these genes were predicted using the online Finder tool (https://www.ncbi.nlm.nih.gov/orffinder, accessed on 8 June 2022). Specific RT-PCR primers (Table 1) were designed to clone and sequence the ORFs. Total RNA was extracted using the HP Total RNA kit (Omega Bio-Tek, Norcross, GA, USA), and first-strand cDNA was generated using HiFiScript cDNA Synthesis Kit (CoWin Biotech, Jiangsu, China). PCR amplification was conducted using a Taq polymerase (Sangon Biotech, Shanghai, China), and amplified products were purified using the DiaSpim^TM^ column DNA gel extraction kit (Sangon Biotech, Shanghai, China) and then sequenced (Tsingke Bio, Chengdu, China). DNAMAN7.0 (Lynnon Biosoft, Vaudreuil, QC, Canada) was used to edit the sequences. ExPASy Molecular Biology Server (https://www.expasy.org/, accessed on 8 June 2022) was used to predict molecular weight and theoretical isoelectric points. GSDS 2.0 (http://gsds.gao-lab.org/, accessed on 8 November 2022) was used to predict the exon–intron organization of MpToll ORFs. Multiple sequence alignments were generated using Clustal X 2.1 and Jalview 2.11 sequence analysis software. Conserved domains were predicted using the SMART tool (http://smart.embl–heidelberg.de/, accessed on 9 November 2022). A phylogenetic tree was performed using the Neighbor-Joining algorithm (1000 replications) with MEGA X software [32].

### 2.3. Analysis of Developmental and Tissue Specificity Expression Patterns

Samples were collected at two time points in the same developmental stage, comprising those just birth or molting (<1 h); those at 12 h of birth or molting for the first-instar, second-instar, third-instar, and fourth-instar nymphs; and wingless and winged adults. None of the nymphs used in this study had wing pads. Tissue samples were collected from 1-day-old wingless adults, including the head, epidermis, gut, hemolymph, and embryo. Three replicates were taken for each stage and tissue. qPCR analysis was performed using the CFX96™ Real-Time Quantitative PCR System (BioRad, Hercules, CA, USA) with FastStart Essential DNA Green Master Mix (Roche, Indianapolis, IN, USA). qPCR primers were designed (Table 1), and the amplification efficiency was evaluated to be between 90% and 120%. *Mpβ-actin* and *MpRps20* were used as internal reference genes. Relative expression was calculated using the 2^−ΔΔCt^ method [33].

### 2.4. Bacterial Challenge

For bacterial infection, examples of gram-negative and -positive bacteria, *Escherichia coli* (strain ATCC25922) and *Staphylococcus aureus* (strain ATCC25923), respectively, were cultured to logarithmic phase at an absorbance (OD_600nm_) of approximately 0.8 in Luria–Bertani (LB) broth at 37 °C. Bacterial cultures were resuspended to 1 × 10^8^ and 1 × 10^9^ colony-forming units/mL for *E. coli* and *S. aureus*, respectively. The 1-day-old wingless adults were anesthetized with CO_2_ and placed on a fluted agar plate (1%), and each adult was injected at the dorsal site of the abdomen with 30 nL of *E. coli* or *S. aureus* solution using an IM-31 microinjector (NARISHIGE, Tokyo, Japan). The control group was injected with an equal volume of sterile water. Injected insects were placed in plastic cups on fresh tobacco leaves and raised in the above climate chamber. Then, 15 surviving individuals were randomly selected from each group at 6, 12, and 24 h post-injection for gene expression analysis, and each treatment was replicated three times.

### 2.5. Functional Analysis Based on RNAi

Specific primers for MpToll genes with a T7 polymerase promoter sequence (Table 1) were designed to synthesize the target dsRNA using a TranscriptAid T7 High Yield Transcription Kit (Thermo Scientific, Waltham, MA, USA). The integrity of the purified dsRNA was determined using a NanoDrop2000 spectrophotometer (Thermo Scientific, Waltham, MA, USA) and 1% agarose gel. To investigate the roles of these genes in *M. persicae* immune response, approximately 250 ng of dsMpToll was injected into fourth-instar nymphs, and the dsRNA of green fluorescent protein (ds*GFP*) was used as a negative control. Then, 15 treated aphids were collected from each group at 24 h and 48 h after injection to determine the RNAi efficiency. Thirty treated aphids were injected with *E. coli* or *S. aureus* suspensions at 24 h after dsRNA treatment from each group to evaluate the susceptibility of *M. persicae*. Control insects were injected with an equal amount of 30 nL of sterile water. The mortality of each group was recorded after 24 h treatment, and three biological replicates were performed for all treatments.

### 2.6. Statistical Analysis

The comparative analyses of all data are presented as mean ± SE (standard error) using SPSS 22.0 (IBM Corp, Chicago, IL, USA). Significant differences between multiple samples were assessed with one-way ANOVA (Tukey’s HSD test), and the comparisons between two samples were analyzed by independent sample Student’s *t*-test (GraphPad Prism 8.0).

## 3. Results

### 3.1. Bioinformatics and Phylogenetic Analysis of MpToll Genes

Identification and sequencing analysis revealed five Toll receptor genes from *M. persicae*: *MpToll*, *MpToll6*, *MpToll6-1*, *MpToll7*, and *MpTollo* (GenBank accession numbers: OP778111, OP778113, OP778114, OP778115, and OP778116, respectively) and included full-length ORF sequences of 2952, 3789, 3882, 4008, and 3747 bp and encoded proteins of 983, 1262, 1293, 1335, and 1248 amino acids, respectively. The predicted molecular weights, theoretical isoelectric points, and instability indices are listed in Table 2. The exon–intron organizations of MpToll genes are shown in Figure 1A. Comparison between genomic and cDNA sequences showed that the coding process of the *MpToll* was interrupted by three introns, whereas none of the other four genes contained introns. Conserved domain analysis indicated that these deduced sequences are members of the Toll receptor family. They all contained a TIR functional domain and a transmembrane domain, as well as an extracellular LRRs domain consisting of 11–25 amino acids, and a signal peptide of 33, 19, 29, 27, and 18 amino acids, respectively (Figure 1B). Multiple sequence alignment identified three conserved motifs in the C-terminal TIR domain of the MpToll proteins: the initial F/YDAxxxxS motif, intermediate CLHYRD motif, and final FWxxL motif. The LRRs domain shares high similarity and contains the characteristic consensus sequence LxxLxLxxNxL (Figure 1C). Phylogenetic analysis of Toll proteins in the evolutionary relationship between *M. persicae* and other insects revealed that the five MpToll proteins were divided into five branches, each of which clustered together with the corresponding Toll proteins of other insects and showed high similarity to those of other aphids (Figure 2).

### 3.2. Developmental and Tissue Expression Profiles of MpToll Genes

Five MpToll genes were continuously expressed in tested stages and tissues, and the expression patterns showed partially similar trends (Figure 3). At just after birth or molting (<1 h), five genes were the lowest expressed in the third-instar nymphs and the highest in first-instar nymphs or winged adults. High expression was also found in wingless adults and fourth-instar nymphs, and low expression was found in second-instar nymphs. At 12 h after birth or molting, the expression patterns of *MpToll* and *MpToll6-1* were similar to those described above. *MpToll6* was the highest expressed in the first-instar nymphs, and there were no significant differences between the other stage. *MpToll7* was highly expressed in the fourth-instar nymphs and adults. *MpTollo* expression was the highest in the first-instar nymphs and the lowest in the third-instar nymphs (Figure 3A). The expressions of these genes were universally highest in the head, followed by the epidermis and embryo, and relatively low expression levels were found in the gut and hemolymph (Figure 3B).

### 3.3. Expression Induced by Bacterial Challenge

The results of induced MpToll genes expression after exposure to the two bacterial species showed that after injection with *E. coli*, the mRNA level of *MpToll6-1* increased continuously at all time points while that of *MpToll7* showed an upward trend at 6 and 12 h. Expression of *MpToll* and *MpToll6* mRNA only increased at 12 and 6 h, respectively. *MpTollo* mRNA expression did not significantly change compared with that in the control group. After *S. aureus* injection, the mRNA expression of *MpToll* and *MpToll6* increased at three time points while that of *MpToll6-1* and *MpTollo* significantly increased at 6 and 12 h, and that of *MpToll7* significantly increased at 6 and 24 h (Figure 4).

### 3.4. Effect of Knockdown of MpToll Genes Expression on Susceptibility to Bacterial Infection

To examine the role of MpToll genes in the immune responses of *M. persicae*, we performed RNAi on all five genes. The transcriptional activities of MpToll genes were detected by qPCR at 24 and 48 h after injection of dsRNA, and the silencing efficiency was assessed. The transcriptional levels of five genes were significantly suppressed by 27–88% and 16–90% at 24 and 48 h after injection of dsMpToll, respectively, compared with that in the control group injected with ds*GFP* (Figure 5A). The susceptibility of *M. persicae* to the two bacterial species showed that the dsMpToll-treated aphids had significantly increased mortality from *E. coli* and *S. aureus* infection (10–54% and 10–58%, respectively) compared with that in the control group injected with sterile water (Figure 5B).

## 4. Discussion

The insect Toll receptor is a key effector of the Toll pathway, recognizing extracellular specific ligands and triggering intracellular cascades to maintain the normal Toll signaling pathway in the immune response and ensure that insects resist pathogen infection and continue normal development [34]. Herein, five Toll receptor genes of *M. persicae* were cloned, which is less than the number of Toll-related genes identified in other insects, ranging from 7 in *Acyrthosiphon pisum* to 14 in *Bombyx mori* [35,36]. Sequence analyses indicated that three typical conserved domains of the Toll family were found in five MpToll amino acid sequences, and a similar conservation has been found in Toll sequences in *B. mori* and *Lymantria dispar* [37,38]. Different domains may play distinct roles, with the TIR domain being a key site of interaction with intracytoplasmic adaptor protein MyD88 to enable intracellular signaling [39], while the LRRs domain was thought to be particularly suitable for interactions between proteins [40]. Phylogenetic analysis showed that the five protein sequences formed a distinct clade and clustered together with the corresponding Tolls from *D. melanogaster*, *B. mori*, *A. gambiae*, and other insects, indicating that Tolls are evolutionarily conserved [37]. These results suggest that these five MpToll genes are typical Toll family members, and their functions may be conserved among the orthologs in each cluster.

The high expression of the MpToll genes was spread in some instars, suggesting that these genes may involve in various biological processes. We found that the five MpToll genes were highly expressed in adults (both wingless and winged) and the fourth-instar nymphs, which is consistent with the expression of *Toll-6* from *Plutella xylostella* [41]. The complex environment of adults and a large amount of feeding by the fourth-instar larvae in the binge-eating stage was speculated to considerably increase the probability of exposure to various pathogens, which was responded to by increasing the expression level of *Toll-6* [41]. We also found that the transcription levels of these five genes were highly expressed in the first-instar nymphs, which may be because of the involvement of Toll in aphid development. Tissue-specific expression analysis of the MpToll genes was the highest in the head suggesting that Toll may have additional functions. The brain is the activation and control center of insect activity, and it is believed that the nervous system is also involved in innate immune response [42]. Studies have shown that Toll signaling is required for the fate determination and differentiation of *Caenorhabditis elegans* neuronal cells, and worms can smell pathogenic bacteria and trigger a behavioral escape response through this mechanism [43]. Toll has been suggested to be involved in embryonic development in *Drosophila* and *A. gambiae* [24,26], and the transcripts of five MpToll genes in *M. persicae* were detected in the embryonic stage in this study. The epidermis is the first line of defense of insect innate immunity, but none of the five MpToll genes were particularly highly expressed in epidermal tissues; however, the epidermis is still thought to play an important defense role in natural immunity [44]. Relatively small amounts of MpToll mRNAs were found in the gut and hemolymph. Similar studies found that *BmToll* was not expressed in the gut of the silkworm, and other family genes, including *BmToll,* were not detected in major immune tissues such as fat body and hemocyte [37].

The Toll pathway is believed to be primarily activated by gram-positive bacteria and fungi, controlling the expression of most antifungal peptides [45]. Ingestion of *E. coli* or *S. aureus* has been found to activate both the Toll and IMD pathways, whereas ingestion of the former was found to activate the expression of most AMP genes while ingestion of the latter induced only some of the AMP genes in *Drosophila* [46]. Recent studies have proposed crosstalk between Toll and Imd pathways in hemipterans and other arthropods [47]. Changes in the expression of Toll receptors have been demonstrated following invasion by various foreign pathogens [48]. In this study, the expression of these five MpToll genes was induced to varying degrees after infection with either of the two bacterial species. Similar results have been found in other insects, such as with the expression of the *P. xylostella Toll-6*, which responded to both gram-negative and -positive bacteria [41]; *E. coli* infection weakly induced the expression of *Toll-1* and *Toll-9* in *A. gambiae* [26]; and *Toll9* in the midgut of silkworm could respond to both *E. coli* and *S. aureus* infection [48]. Therefore, we suggest that these MpToll genes may be involved in the immunomodulatory process of *M. persicae* against different bacteria.

In the process of survival and reproduction, the innate immune system of aphids plays an important role in resisting pathogen invasion. Recent studies have shown that the suppression of the key gene of the innate immune system led to the increased mortality of *A. pisum* upon infection by bacteria [8,30]. Herein, silencing of the five MpToll genes significantly increased the mortality of *M. persicae* adults after infections with *E. coli* or *S. aureus*, further suggesting that these MpToll genes are crucial for *M. persicae* to resist bacterial infection. Innate immunity is present in all organisms and is evolutionarily conserved [49]. Immune-related proteins such as AMPs and PGRPs have been extensively studied in fruit flies, beetles, and other insects [46,50]. The production of AMPs was initially believed to be a common feature of insect immune response. However, a later study indicated that infecting *A. pisum* with gram-positive *Micrococcus luteus* caused lysozyme-like activity to be reduced and did not detect any AMP homologues [51]. Subsequent annotation of the immune and stress gene banks of *A. pisum* found that the genome lacked a large number of key genes in the IMD signaling pathway and did not contain AMP genes ubiquitous in other insects [35]. Our results suggest that the Toll receptors are essential for *M. persicae* to resist bacterial infection. However, the Toll pathway mechanism in aphid immunity remains unclear, and further research is needed to clarify these complex processes.

## 5. Conclusions

We identified five ORFs encoding Toll receptors with typical conserved domains of the Toll family in *M. persicae*. Our study elucidated the structure, phylogenetic relationship, and spatiotemporal expression patterns of these MpToll genes. Furthermore, bacteria-induced expression and RNAi results demonstrate that MpToll genes play an important role in the response of *M. persicae* to bacterial stress, which further enriches immune knowledge of aphids and provides a reference for the search for control targets of aphids in the future.

## Figures and Tables

**Figure 1 insects-14-00275-f001:**
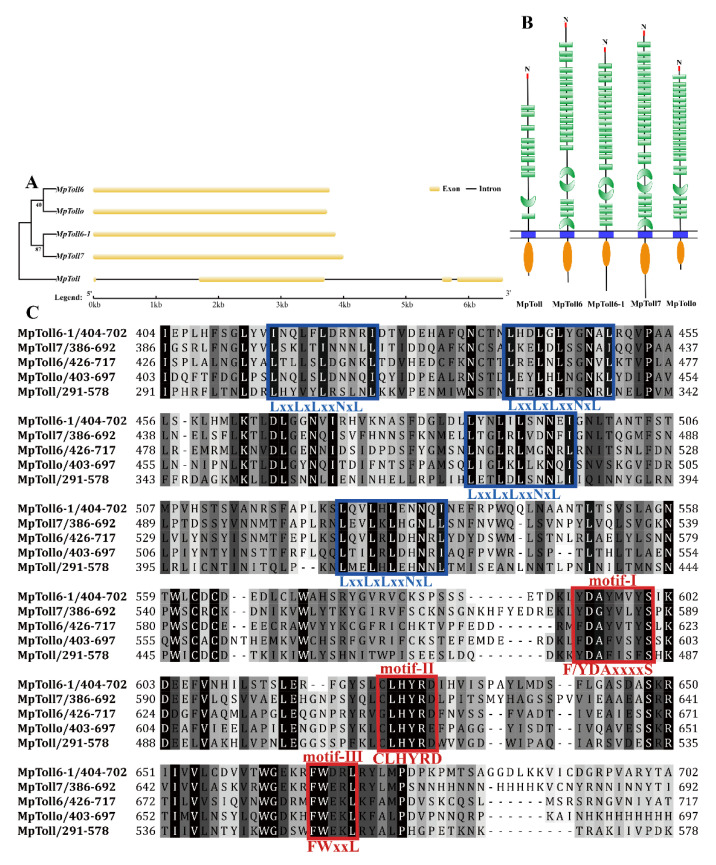
(**A**) Exon–intron organization in MpToll ORFs. The exons are shown in pale yellow rectangle boxes, and the black lines between boxes indicate the introns. (**B**) Conserved domains of MpToll proteins. The red stripes indicate signal peptides, green bars and horseshoe shapes indicate the LRRs domain, blue bars indicate the transmembrane domain, and orange shapes indicate the TIR domain. (**C**) Multiple sequence alignment of MpToll proteins. The red frame refers to the three important conserved motifs of the TIR domain, and blue frame refers to the feature consistency sequence of the extracellular LRRs domain.

**Figure 2 insects-14-00275-f002:**
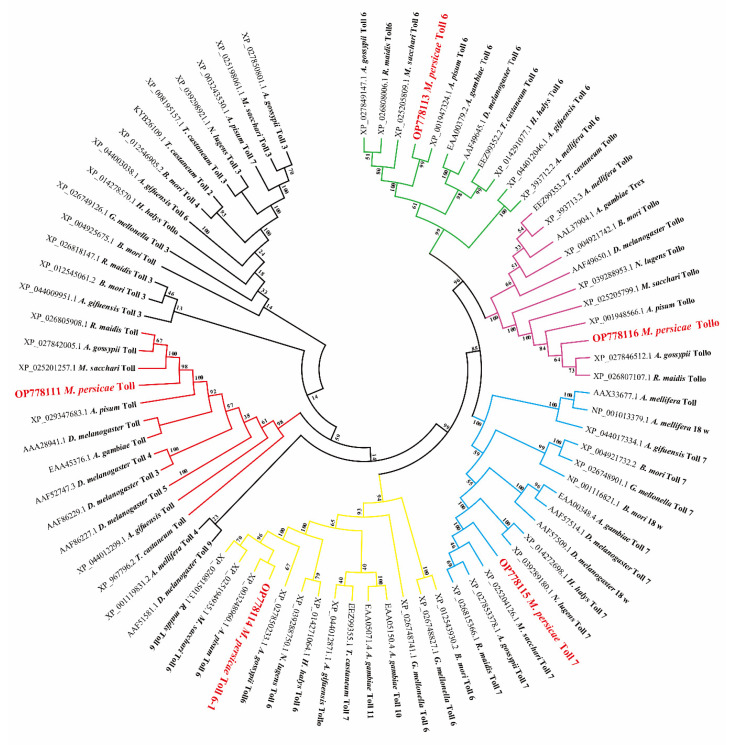
Phylogenetic analysis of insect Toll proteins. The MpToll sequences were labeled with a red font and bold typeface. Insect species include *Myzus persicae*, *Acyrthosiphon pisum*, *Aphis gossypii*, *Rhopalosiphum maidis*, *Nilaparvata lugens*, *Drosophila melanogaster*, *Tribolium castaneum*, *Bombyx mori*, *Halyomorpha halys*, *Melanaphis sacchari*, *Galleria mellonella*, *Anopheles gambiae*, *Apis mellifera,* and *Aphidius gifuensis*. The GenBank accession number of each species is listed in the tree.

**Figure 3 insects-14-00275-f003:**
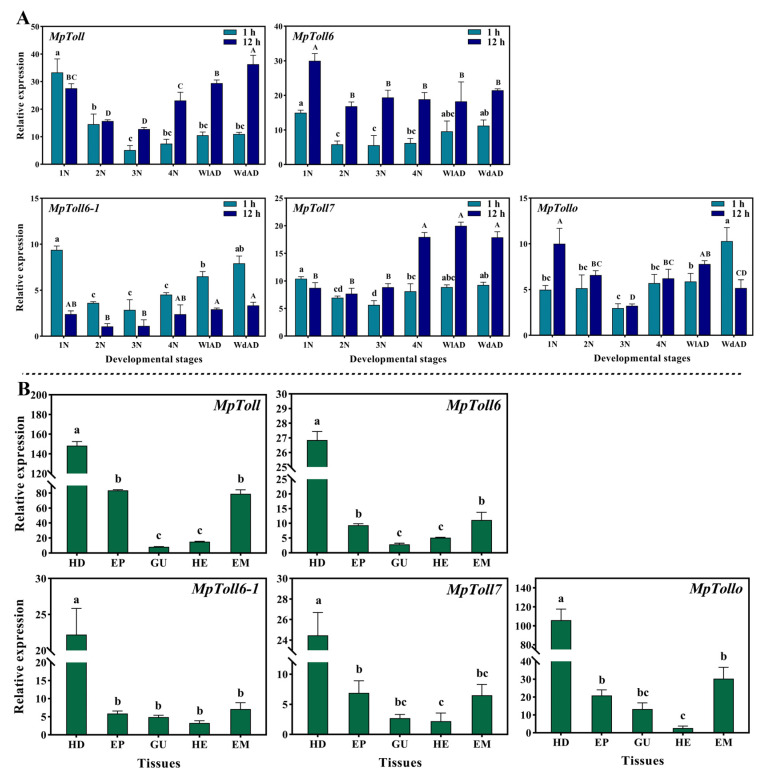
(**A**) Specific expression profiles of MpToll genes at different developmental stages. The peacock blue bar indicated the relative expression just birth or molting (<1 h) for the corresponding instar, and the dark blue is 12 h after birth or molting. Significant differences are shown in lowercase letters for the former and in capital letters for the latter (*p* < 0.05, Tukey). (**B**) Specific expression profiles of the five MpToll genes in different tissues of adults; the letters above the bars indicate significant differences (*p* < 0.05, Tukey). Developmental stages: first-instar (1N), second-instar (2N), third-instar (3N), and fourth-instar (4N) nymphs, wingless (WlAD), and winged (WdAD) adults. Tissue samples: head (HD), epidermis (EP), gut (GU), hemolymph (HE), and embryos (EM) from adults.

**Figure 4 insects-14-00275-f004:**
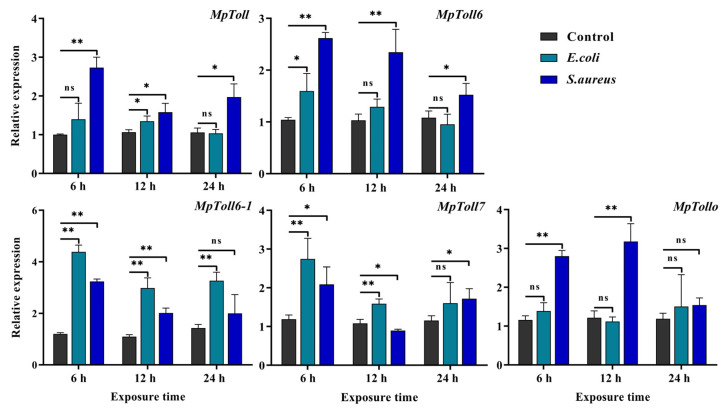
The mRNA expression profiles of MpToll genes after challenge with *E. coli* or *S. aureus*. Sterile water group was used as a control. Significant differences between the treatment and control groups were determined using Student’s *t*-test (*, *p* < 0.05, **, *p* < 0.01).

**Figure 5 insects-14-00275-f005:**
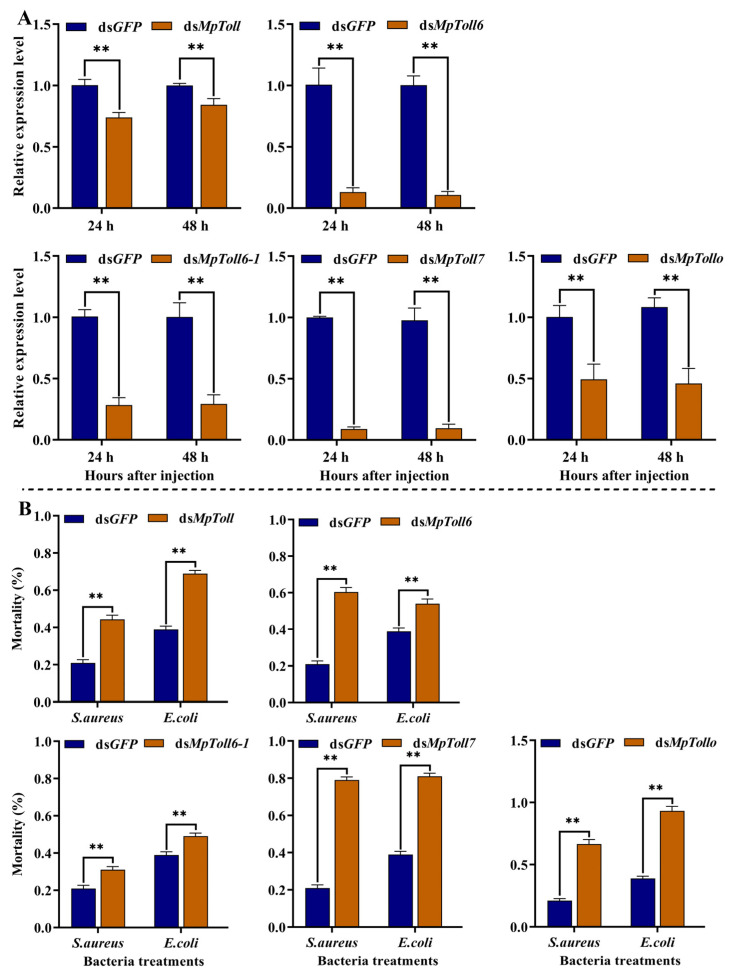
(**A**) Silence efficiency of MpToll genes expression in the fourth-instar nymphs of *M. persicae* by RNAi. (**B**) Mortality of *M. persicae* 24 h after injection of *E. coli* or *S. aureus* (48 h after injection of dsRNA). Asterisks indicate the differences between the experimental group and the ds*GFP* group (Student’s *t*-test, **, *p* < 0.01).

**Table 1 insects-14-00275-t001:** Primers used in this study.

Application of Primers	Primer Name	Forward Primer (5′–3′)	Reverse Primer (5′–3′)
Genes cloned	*MpToll* F1/R1	TTCGTGTTCGCAGTTGTC	GCTTGGTCAGGCATAGTAA
	*MpToll* F2/R2	GGCGTCTCAATCAACTACT	TCAGTCTTGTTGTCATCCTT
	*MpToll* F3/R3	GGATGACAACAAGACTGATT	GTGGAAGGGCATATCTCAA
	*MpToll6* F1/R1	GCGAGTATGTGCGTGTTCTT	CGGCTTATCGTTGACCACTG
	*MpToll6* F2/R2	TGCCGTAATGCCGTCCAA	CAGCCTGATCGCTTCTATGC
	*MpToll6* F3/R3	ATCGGCTCCGCAATATAACTTC	CTCGCAATCACACGCATCAA
	*MpToll6* F4/R4	GATGCGTGTGATTGCGAGAT	ACCGTCATCTTTCAGGCTGTA
	*MpToll6* F5/R5	GGACGTAAGTATTCGAGACCAA	GCAAATCACGCCCAAAGATG
	*MpToll6-1* F1/R1	AGTATTGCGAGTGTCCGTCTT	AGGTTCAGGTTGGCGAGTC
	*MpToll6-1* F2/R2	CGACAACAACATCTGGAACCT	CGAGCGTCTTGAGCATGTG
	*MpToll6-1* F3/R3	TCTTGGACAGGAACCGCATC	CTCGTACTCGCACAGGAACT
	*MpToll6-1* F4/R4	CCGCATTATGGACATGGACTC	TACGAACACGACGATGAACAG
	*MpToll6-1* F5/R5	TGCTGTGCTCGGACAAGAT	ACAGACAAGAAGAAGCAGATGG
	*MpToll7* F1/R1	ATTATTGTCGGCTACTCGG	ATCTCGGTGATGCTGTTG
	*MpToll7* F2/R2	GCACACGTCCAACTTAGA	TTGAGTCAGGTTACCGATAA
	*MpToll7* F3/R3	CGGATTGCGACTTGTAGA	AACGAGCTAAGCCTGTTG
	*MpToll7* F4/R4	CATCGTGGACTGTTCGTA	CACTCGGTCTGTATGAAGT
	*MpToll7* F5/R5	GCAACAAACACTTCTACGAGGA	GTAACGACGACTTGCGAGG
	*MpTollo* F1/R1	TGCCCGTTCGTTGATGTCT	AAGTTGGTCAGGTGCGAGAA
	*MpTollo* F2/R2	TTCTCGCACCTGACCAACTT	AATAATTGCCGTCCACCCTGA
	*MpTollo* F3/R3	CATCTCATCAACCTCGCCAATA	TGGAACTGACGCTGTGGAA
	*MpTollo* F4/R4	AAGAAGCGTCCGAAGAACAG	TATATGAATGGCAAGCGGTCTT
qPCR (q)	q*MpToll* F/R	AAGGATGACAACAAGACTGA	GGCAACTTGGTGATGGAA
	q*MpToll6* F/R	CTTATTAGGCTGGTTGCGTTGA	CGAGGTTCCGAAGTGGTATGA
	q*MpToll6-1* F/R	ACTGTTCCATTCCACTCGGTAT	AGGTGCTGGTCAACTCGTT
	q*MpToll7* F/R	ATACCGAACAGCGTGGAAGT	TTAACTCGTTGGCGTACAAGTC
	q*MpTollo* F/R	CGGTCGTCACTCACAACAAC	CGGCAACTTGGCGATGTT
	q*Mpβ-actin* F/R	TGGTATCGTCTTGGATTCTG	TTAGGTAGTCGGTGAGATCA
	q*MpRps20* F/R	CTCCGAGATGATTGCCGATAT	GTCTTTGAACCTTCACCACAAG
dsRNA synthesis (ds)	ds*MpToll* F/R	T7-CGTGCCATCAATCATCAACTCT	T7-AGGAATCCAATCGCCGACTAC
	ds*MpToll6* F/R	T7-GCCTTCGTCCAGTCAGTGT	T7-CAACGCAACCAGCCTAATAAGA
	ds*MpToll6-1* F/R	T7-ACTGTTCCATTCCACTCGGTAT	T7-GATGCGGTTCCTGTCCAAGA
	ds*MpToll7* F/R	T7-TGCCAGTCACAATCACATAACC	T7-GCGTCGTATTCACAACACTTG
	ds*MpTollo* F/R	T7-TTCCACAGCGTCAGTTCCA	T7-GCGAGCGTCAGATGAGTCA
	ds*GFP* F/R	T7-GCCAACACTTGTCACTACTT	T7-GGAGTATTTTGTTGATAATGGTCG

Note: T7-, T7 promoter sequence TAATACGACTCACTATAGGG; qPCR, real-time quantitative polymerase chain reaction; dsRNA, double-stranded RNA.

**Table 2 insects-14-00275-t002:** Bioinformatic characterization of MpToll genes.

Gene	ORF Length (bp)	Protein Sequence (aa)	Molecular Weight (kDa)	Isoelectric Points	Leucine Ratio (%)	Instability Index
*MpToll*	2952	983	113.69	6.42	15.3 (150)	42.45
*MpToll6*	3789	1262	142.50	5.54	12.4 (157)	45.48
*MpToll6-1*	3882	1293	146.24	5.89	13.9 (180)	40.65
*MpToll7*	4008	1335	151.83	6.80	12.7 (170)	43.18
*MpTollo*	3747	1248	142.14	6.07	13.3 (166)	36.35

## Data Availability

All data are provided within the text.

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
