# Peer review of "Characterization and Functional Analysis of Toll Receptor Genes during Antibacterial Immunity in the Green Peach Aphid *Myzus persicae* (Sulzer)"

_insects, 2023, doi:10.3390/insects14030275_

Round 1

Reviewer 1 Report

The manuscript describes the identification of toll genes in the aphid, Myzus Persicae, and the analyses of changes in expression (temporal and spatial), induction of expression, phenotypes of RNAi. The data can be predictable fro the past studies and may not be surprising, but the analyses and descriptions are very concrete and therefore are worth being published as a paper of analysis for the new species. By reading this manuscript, the reader may be able to follow correctly the history of the study on toll-related things. 

the below are minor comments:

line 63: different insects, including …

can be changed to “wide range of insect orders, including”

line 64: after “in mammal [21].” is it possible to add a sentence, “It is interesting that toll-like receptors in the Atlantic cod, Gadus morhua, have been diversified for compensation of their deficiency of adaptive immunity.” ?

line 110:

were the exon-intron prediction done also by alignment of cDNA and genome sequences?

line 121-123:

just a question from my curiosity, but was it difficult to take tissues from small insects like aphids?

line 174:

is this the same in other species/

line 178-179:

are the signal sequences for secretion or targeted to membrane?

line 206-217:

can the description be shortened?

line 238:

It is possible that injury by injection could leads to increase of expression levels, therefore ideally it may be good to analyze also naive samples or those from 0 time point. but at least, the data show significant difference between control and infected samples.

line 275:

“and may function as cell membrane receptors to transmit signals ” can be

“and also their functions may be conserved among the orthologs in each cluster. “?

line 305:

is it good to add a sentence of “, > the recent studies have proposed crosstalk between Toll and other pathways including Imd pathway in several speceis (please add references)”, after “AMP genes in Drosophila \i0[45]. ” ?

Author Response

Thank you very much for the valuable comments of the reviewer!

Point 1: line 63: different insects, including …

can be changed to “wide range of insect orders, including”

Response 1: I've rephrased the sentence.

Point 2: line 64: after “in mammal [21].” is it possible to add a sentence, “It is interesting that toll-like receptors in the Atlantic cod, Gadus morhua, have been diversified for compensation of their deficiency of adaptive immunity.”?

Response 2: I've added the sentence, and added a reference support.

Point 3: line 110:

were the exon-intron prediction done also by alignment of cDNA and genome sequences?

Response 3: Yes, we obtained annotation information after comparing the target cDNA sequence with the genome sequence (PRJNA296778), and mapped it using GSDS 2.0 software (http://gsds.gao-lab.org/) to get our results.

Point 4: line 121-123:

just a question from my curiosity, but was it difficult to take tissues from small insects like aphids?

Response 4: Due to the small size of the aphids, it was indeed difficult to extract the intact tissue at the beginning, but after mastering the anatomical method, the tissue can be extracted smoothly.

Point 5: line 174:

is this the same in other species/

Response 5: Similar results have been found in studies of Anopheles gambiae. Here is the reference:

Luna, C.; Wang, X.; Huang, Y.; Zhang, J.; Zheng, L. Characterization of four Toll related genes during development and immune responses in Anopheles gambiae. Insect Biochem Mol Biol 2002, 32, 1171–1179.

Point 6: line 178-179:

are the signal sequences for secretion or targeted to membrane?

Response 6: I've corrected this mistake.

Point 7: line 206-217:

can the description be shortened?

Response 7: I've made appropriate adjustments to this part.

Point 8: line 238:

It is possible that injury by injection could leads to increase of expression levels, therefore ideally it may be good to analyze also naive samples or those from 0 time point. but at least, the data show significant difference between control and infected samples.

Response 8: It is indeed a pity that no analysis of naive samples or those from 0 time point, but we were able to eliminate the effects of injectable injury by setting up control.

Point 9: line 275:

“and may function as cell membrane receptors to transmit signals ” can be

“and also their functions may be conserved among the orthologs in each cluster. ”?

Response 9: I've rephrased the sentence.

Point 10: line 305:

is it good to add a sentence of “, > the recent studies have proposed crosstalk between Toll and other pathways including Imd pathway in several speceis (please add references)”, after “AMP genes in Drosophila \i0[45]. ” ?

Response 10: I've added the sentence, and added a reference support.

Thank you again for your valuable comments, I have carefully revised the whole text according to your advice.

Reviewer 2 Report

The draft of article number 2229521 submitted to the Insects, MDPI, entitled” Characterization and functional analysis of Toll receptor genes during antibacterial immunity in the green peach aphid Myzus persicae (Sulzer)” carried results in the text that needs revision for the improvement of the draft. Some suggested changes for example are in the comments portion to revise and improve the manuscript. Please find suggested corrections and journal-style format for revision. 

Line 39: JNK, and JAK/STAT” please explain the abbreviations where first used

Line 42:  Toll/Toll-like receptors” what is this please correct and or recheck

Line 116: T7 TAATACGACTCACTATAGGG” what is this sequence and why this sequence is out of the table

Line 132: S aureus” please write the complete scientific name

Line 150: injected into fourth-instar nymphs” why fourth instar nymph and no other instars

Line 177-178: Proteins contained signal peptides of 33, 19, 29, 29, and 18 amino acids” please recheck if the figures are correctly written.

Line 180-183: Please explain the abbreviations where first used

Table 2: Please edit and format the table according to the author’s instructions and also explain the abbreviations used in the table

Figure IC:  Result visibility of the sequence comparison should be clearer for the readers

Figure 2. Phylogenetic tree of Toll proteins from Acyrthosiphon pisum, Aphis gossypii, Rhopalosiphum  maidis, Nilaparvata lugens, Drosophila melanogaster, Tribolium castaneum, Bombyx mori, Halyomorpha  halys, Melanaphis sacchari, Galleria mellonella, Anopheles gambiae, Apis mellifera, Aphidius gifuensis, and  M. persicae. The MpToll sequences were labeled with a red font and bold typeface and the GenBank accession number of each species is listed in the tree.” Rephrase the title of the table and better to place the species used in the supplementary data if necessary

207-208: genes were generally expressed at lower levels in the third- instar nymph and at relatively higher levels in the first-instar nymphs as well as in adult stages” what’s about the results of 2nd instars.

Line 276-277: The specific expression of the MpToll genes was widely spread in all instars, suggesting that these genes are involved in various developmental processes” Please rephrase the sentence. These results are from the present study or suitable reference” please also explain which developmental processes.

Line 271-273: Phylogenetic analysis showed that the five  protein sequences formed a distinct clade and clustered together with the corresponding Tolls from D. melanogaster, B. mori, A. gambiae, and other insects”, Please add the suitable reference

Please double-check for inconsistencies in Journal style/formatting/ authors instructions, double spaces, spellings of the words, English vocabulary, missing italics, scientific names, excessive/missing information, etc.

Author Response

Thank you very much for the valuable comments of the reviewer!

Point 1:

Line 39: JNK, and JAK/STAT” please explain the abbreviations where first used

Response 1: I have added the explanation in the paper.

Point 2: Line 42: Toll/Toll-like receptors” what is this please correct and or recheck

Response 2: I've corrected my expression.

Point 3: Line 116: T7 TAATACGACTCACTATAGGG” what is this sequence and why this sequence is out of the table

Response 3: I've explained this sequence and annotated the table. Since adding this T7 promoter sequence would make the overall primer sequences too long, which would affect the beauty of the table, I put it in the notes.

Point 4: Line 132: S aureus” please write the complete scientific name

Response 4: I've written out the complete scientific name.

Point 5: Line 150: injected into fourth-instar nymphs” why fourth instar nymph and no other instars

Response 5: According to the results of our experiment (figure 3A), the best dsRNA injection stage should be the third-instar nymphs. However, due to the small body size of the Myzus persicae, we found that the mortality caused by mechanical damage was too high during the injection process of the third-instar nymphs, so we chose the fourth-instar nymphs as the object of dsRNA injection.

Point 6: Line 177-178: Proteins contained signal peptides of 33, 19, 29, 29, and 18 amino acids” please recheck if the figures are correctly written.

Response 6: I've checked the writing of the figures and modified it.

Point 7: Line 180-183: Please explain the abbreviations where first used

Response 7: I've explained the abbreviations here in the introduction, line 65.

Point 8: Table 2: Please edit and format the table according to the author’s instructions and also explain the abbreviations used in the table

Response 8: I've modified the table format and explained the abbreviations in the table.

Point 9: Figure IC: Result visibility of the sequence comparison should be clearer for the readers

Response 9: I've adjusted the figure 1C.

Point 10: Figure 2. Phylogenetic tree of Toll proteins from Acyrthosiphon pisum, Aphis gossypii, Rhopalosiphum maidis, Nilaparvata lugens, Drosophila melanogaster, Tribolium castaneum, Bombyx mori, Halyomorpha halys, Melanaphis sacchari, Galleria mellonella, Anopheles gambiae, Apis mellifera, Aphidius gifuensis, and M. persicae. The MpToll sequences were labeled with a red font and bold typeface and the GenBank accession number of each species is listed in the tree.” Rephrase the title of the table and better to place the species used in the supplementary data if necessary

Response 10: I've rephrased the title of the table. The GenBank accession number of each species is listed in the phylogenetic tree, and the species names used can also be placed directly in the figure information, and do not need to be used as supplementary data.

Point 11: 207-208: genes were generally expressed at lower levels in the third- instar nymph and at relatively higher levels in the first-instar nymphs as well as in adult stages” what’s about the results of 2nd instars.

Response 11: I've redescribed the results of the experiment.

Point 12: Line 276-277: The specific expression of the MpToll genes was widely spread in all instars, suggesting that these genes are involved in various developmental processes” Please rephrase the sentence. These results are from the present study or suitable reference” please also explain which developmental processes.

Response 12: I've rephrased the sentence. We analyzed the expression of five genes in different developmental stages of the M. persicae and found the high expression of these genes were spread in some instars (figure 3A), from this we speculate that they may be involved in the development and immune processes of M. persicae, which was also discussed in the paper.

Point 13: Line 271-273: Phylogenetic analysis showed that the five protein sequences formed a distinct clade and clustered together with the corresponding Tolls from D. melanogaster, B. mori, A. gambiae, and other insects”, Please add the suitable reference

Response 13: I've added a reference support.

Point 14: Please double-check for inconsistencies in Journal style/formatting/ authors instructions, double spaces, spellings of the words, English vocabulary, missing italics, scientific names, excessive/missing information, etc.

Response 14: Thank you again for your valuable comments, I have carefully revised the whole text according to your advice.

Round 2

Reviewer 2 Report

The draft of article number 2229521 submitted to the Insects, MDPI, entitled” Characterization and functional analysis of Toll receptor genes during antibacterial immunity in the green peach aphid Myzus persicae (Sulzer)” carried interesting results and the draft has already improved after revision. 

Author Response

Dear Reviewer,

Thank you again for your valuable comments, the quality of the manuscript has been improved after modification according to your suggestions. Wish you good health and all the best!

With best regards,

HeLi